# Timing of Ureteric Stent Removal and Occurrence of Urological Complications after Kidney Transplantation: A Systematic Review and Meta-Analysis

**DOI:** 10.3390/jcm8050689

**Published:** 2019-05-16

**Authors:** Isis J. Visser, Jasper P. T. van der Staaij, Anand Muthusamy, Michelle Willicombe, Jeffrey A. Lafranca, Frank J. M. F. Dor

**Affiliations:** 1Imperial College Renal and Transplant Centre, Imperial College NHS Healthcare Trust, Hammersmith Hospital, London W12 0HS, UK; isisjasmijn@hotmail.com (I.J.V.); jaspervanderstaay@hotmail.com (J.P.T.v.d.S.); anand.muthusamy@nhs.net (A.M.); michelle.willicombe@nhs.net (M.W.); j.a.lafranca@gmail.com (J.A.L.); 2Department of Surgery and Cancer, Imperial College, London W12 0HS, UK

**Keywords:** kidney transplantation, urological complications, ureteric stent, urinary tract infection, timing of removal

## Abstract

Implanting a ureteric stent during ureteroneocystostomy reduces the risk of leakage and ureteral stenosis after kidney transplantation (KTx), but it may also predispose to urinary tract infections (UTIs). The aim of this study is to determine the optimal timing for ureteric stent removal after KTx. Searches were performed in EMBASE, MEDLINE Ovid, Cochrane CENTRAL, Web of Science, and Google Scholar (until November 2017). For this systematic review, all aspects of the Cochrane Handbook for Interventional Systematic Reviews were followed and it was written based on the PRISMA-statement. Articles discussing JJ-stents (double-J stents) and their time of removal in relation to outcomes, UTIs, urinary leakage, ureteral stenosis or reintervention were included. One-thousand-and-forty-three articles were identified, of which fourteen articles (three randomised controlled trials, nine retrospective cohort studies, and two prospective cohort studies) were included (describing in total *n* = 3612 patients). Meta-analysis using random effect models showed a significant reduction of UTIs when stents were removed earlier than three weeks (OR 0.49, CI 95%, 0.33 to 0.75, *p* = 0.0009). Regarding incidence of urinary leakage, there was no significant difference between early (<3 weeks) and late stent removal (>3 weeks) (OR 0.60, CI 95%, 0.29 to 1.23, *p* = 0.16). Based on our results, earlier stent removal (<3 weeks) was associated with a decreased incidence of UTIs and did not show a higher incidence of urinary leakage compared to later removal (>3 weeks). We recommend that the routine removal of ureteric stents implanted during KTx should be performed around three weeks post-operatively.

## 1. Introduction

Kidney transplantation (KTx) is considered the best option for end-stage renal disease (ESRD) management [1,2]. Kidney transplantation increases the life expectancy and quality of life of ESRD patients significantly compared to dialysis [3]. However, KTx is not without peri-operative complication risks; urinary tract infections (UTIs), urinary leakage, and ureteral stenosis are the most frequently seen urological complications. These complications are likely to compromise graft functions [4,5,6,7]. In order to minimize leakage and stenosis, in general, a stent is inserted in the ureter during implantation. Ureteric stents decrease the risk of these urological complications by five to ten-fold. [6,8,9] Most centres use a variation of a JJ-stent (double-J stent), with the typical pigtail end preventing stent migration by positioning one end in the pyelum. The other end remains in the bladder after the ureteroneocystostomy (UNC) is created. The stent can easily be removed by flexible cystoscopy later. Various randomised controlled trials demonstrate that JJ-stenting reduces urinary leakage and ureteral stenosis [5,10,11,12]. Meta-analyses by Wilson and Mangus [6,9] also confirmed these results. Although the use of a JJ-stent reduces the risk of urinary leakage and ureteral stenosis, it may also predispose to UTIs [12,13,14,15,16]. In the existing literature, there is no consensus about the preferred time of stent removal. The European Association of Urology states in its renal transplantation guideline that stent retention longer than 30 days is associated with an increased risk (6% versus 40%) of UTIs [12,17]. Therefore, the guideline advises stents to be removed earlier than six weeks post-transplant (which is protocol in most transplant centres) rather than later. Therefore, in the last two years, studies have started to investigate different timings of stent removal within one-month post-transplant [8,18,19,20,21,22,23]. Recently, Thompson et al. [24] published a review regarding this topic; however, the authors included fewer studies and did provide a robust conclusion about timing of stent removal as their focus was more on the difference between per-urethral and bladder indwelling stents. 

The aim of this systematic review is to give a comprehensive overview of currently available literature and to investigate if meta-analysis can elucidate whether a more definite timing for stent removal can be determined.

## 2. Methods

All aspects of the Cochrane Handbook for Interventional Systematic Reviews were followed, and the study was written according to the Preferred Reporting Items for Systematic Reviews and Meta-Analyses (PRISMA) statement [25,26]. Details of the protocol for this systematic review were registered on PROSPERO (ID: CRD42018079867) and can be accessed online [27].

### 2.1. Literature Search Strategy

A literature search for all articles regarding JJ-stenting after KTx was performed in EMBASE, MEDLINE Ovid, CENTRAL (the Cochrane Library 2017, issue 11), Web of Science, and Google Scholar. The search was performed for articles published up to November 2017 relevant to outcomes of ureteric stent placement and timings of removal. The search strings for each respective database are attached in Appendix A. Reference lists of the identified relevant articles were manually scrutinized to ensure that no articles were missed.

### 2.2. Literature Screening

Study selection was performed independently by two authors (J.P.T.v.d.S. and I.J.V.). Study inclusion was carried out in two phases: after an initial title and abstract selection, full articles of the abstracts regarded as potentially eligible were retrieved and underwent complete review and assessment until a final inclusion was made. When a discrepancy in inclusion between the two authors occurred, articles were discussed with two senior authors (J.A.L., F.J.M.F.D.) in order to reach a consensus.

### 2.3. Data Extraction

Studies were assessed for timing of stent removal and the incidence of UTIs, urinary leakage, ureteral stenosis, and reintervention. Other parameters that were assessed were donor type, mean recipient age, the type of stent, technique of UNC, technique of stent removal, immunosuppressive therapy and antibiotic prophylaxis regimen. When the type of stent was not specified in a particular article, we reached out to the authors. If the authors did not respond, we noted the stent-type as “unspecified” but did not mark it as an exclusion criterium. 

Studies were only included if they indicated time of stent removal and at least one of the following outcome parameters: UTI, urinary leakage, and/or ureteral stenosis. If a study stated an outcome as “major urological complication” (MUC) and we were not able to define whether this included stenosis or leakage, we analysed this outcome parameter separately as MUC.

To define UTI, we used the Guideline for Urological Infections of The European Association of Urology [28]. It states that a positive urine culture with a bacterial colony count of more than 10^5^ colony-forming units per mL urine is defined as a UTI. However, if patients had lower counts of colony-forming units per mL urine but were reported to have symptoms of a UTI, we chose not to exclude them. When authors of the articles did not define a UTI, we assumed that the official definition was used.

We did not find an official guideline defining urinary leakage and ureteral stenosis. However, Dominguez et al. [10] stated the following definitions for urinary leakage and ureteral stenosis: leakage is defined as drainage or accumulation of perirenal fluid with characteristics of urine and ureteral stenosis is defined as impairment of adequate kidney drainage demonstrated at ultrasound (US) or intravenous pyelogram. When authors of the articles did not define urinary leakage and/or ureteral stenosis, we assumed the abovementioned definitions.

### 2.4. Critical Appraisal

The level of evidence of each selected paper was established using the GRADE tool [29]. The GRADE approach defines the quality of a body of evidence by consideration of within study risk of bias (methodological quality), directness of evidence, heterogeneity, precision of effect estimates, and risk of publication bias. 

### 2.5. Statistical Analysis

Articles were assessed as to whether they were suitable for quantitative analysis. If articles compared two or more groups with different timings of stent removal and if those groups could be divided in an early and late timing of stent removal with a cut off at three weeks, they were included for meta-analysis. 

Review Manager Software (RevMan 5.3; The Nordic Cochrane Centre, Copenhagen, Denmark) was used for meta-analysis [30]. Each study was weighted by sample size. Heterogeneity of time of stent removal effects between studies was tested using the Q (heterogeneity χ^2^) and the *I*^2^ statistics. A random effects model was used for calculating the summary estimates (odds ratio (OR)) and 95% confidence intervals (CIs) to account for possible heterogeneity. Overall, the effects were determined using a Z-test. In addition, sensitivity analyses were performed to examine whether removing a particular study would significantly change the results and were presented in funnel plots.

## 3. Results

Of the 1043 articles identified from the initial literature search, fourteen articles were within the scope of this systematic review; three randomised controlled trials (RCTs) [20,21,31], nine retrospective cohort studies [8,13,19,22,23,32,33,34,35] and two prospective studies [36,37]. A total of 3216 patients were included, of which 2406 patients (74.8%) underwent living donor KTx (two studies did not record if they used living or deceased donors [34,35]). Table 1 presents an overview of the included studies. In Figure 1, the PRISMA flowchart diagram for systematic reviews is presented. The quality assessment of the included studies is depicted in a Summary of Findings table in Figure 2.

We could not specify the type of stent in five studies. The authors of these five papers were contacted, but unfortunately, we received no response [19,34,35,36,37]. All studies reported the age of the recipients, except for the preliminary results in the three included abstracts [34,35,36]. Only four studies reported that they both included adults and children [19,21,23,37]. All articles described the incidence of UTIs, and nine articles also reported urinary leakage and/or ureteral stenosis [19,20,21,31,32,33,35,36,37]. Table 2 presents the incidence of the different outcome parameters for each study. Regarding the used UNC technique, seven out of the fourteen articles described the use of the Lich–Gregoir technique [13,20,21,22,23,31,32]. Seven articles did not specify which surgical technique was used [8,19,33,34,35,36,37]. Regarding stent removal, ten articles used a cystoscopy for removal [8,20,21,22,23,32,33,34,35,36]. In four studies, the stent was removed along with the urinary catheter seven days after transplantation [21,34,35,36]. Two studies removed the stent on the seventh day posttransplant by pulling on strings that were placed around the stent at the time of transplantation [8,20]. Four studies did not describe the technique of stent removal [13,31,37,38]. Ten studies reported the use of antibiotic prophylaxis to protect against UTI after KTx, of which seven studies also reported the duration of antibiotic prophylaxis regimen [13,20,21,23,31,32,33]. Six studies specified the use of co-trimoxazole as prophylaxis [13,21,22,23,31,32]. Four studies did not record the use of prophylactic antibiotics at all [8,36,37,38]. Seven studies reported which immunosuppressive regimen KTx recipients received [8,20,22,23,31,32,33]. An overview of all the characteristics of the included studies can be found in Table A1 in Appendix A. Four studies could not be included in the meta-analysis because the timing of stent removal was much earlier (e.g., earlier and later than four days) or much later (e.g., earlier and later than seven weeks) than the cut-off value of three weeks [18,39,40,41].

### 3.1. Urinary Tract Infection

Fourteen studies described the incidence of UTIs with different timings of stent removal and were included for meta-analysis (a total of 3216 patients). There was a significant difference between the groups in the risk of developing UTIs, favouring early stent removal (OR 0.49, CI 95%, 0.33 to 0.75 *p* = 0.0009) (Figure 3). Sensitivity analysis showed no change of significance. (Figure A1, Appendix A). 

### 3.2. Urinary Leakage

Eight studies described the incidence of urinary leakage: three RCTs [20,21,31], one prospective study [37] and four retrospective studies [32,33,35,38]. One of these studies described zero events of urinary leakage; therefore, seven studies remained for meta-analysis, with a total of 1505 patients [21,31,32,33,35,37,38]. After pooling the data, there was no significant difference between groups in the risk of developing urinary leakage (OR 0.60, CI 95%, 0.29 to 1.23, *p* = 0.16) (Figure 4). Sensitivity analysis showed no change in significance (Figure A2, Appendix A).

### 3.3. Ureteral Stenosis

Five studies described the incidence of ureteral stenosis [20,21,32,33,36]. Three out of these seven studies reported zero incidents of ureteral stenosis in both groups [20,32,33]. Patel et al. [21] described one case of ureteral stenosis in both the early and late group of stent removal (1.2% and 0.8%, respectively). Gunawansa et al. [36] reported two cases of ureteral stenosis in the late stent removal group (1.1%). No meta-analysis was performed given the low incidence of ureteral stenosis. 

Dadkhah et al. [37] and Asgari et al. [19] recorded the incidence of hydronephrosis; however, they did not describe the cause of the hydronephrosis. Dadkhah et al. [37] reported eleven cases in the early stent removal group (3.4%) versus three (2.8%) in the late group of stent removal (*p* = 0.122). Asgari et al. [19] reported, respectively, seven (11.5%) and four (13.3%) cases in the early and late group of stent removal (*p* = 0.71). 

Some studies only reported MUC without defining whether this was urinary leakage or ureteral stenosis [13,23,34]. We decided to perform an additional meta-analysis of MUC. We included data from those studies and combined ureteral stenosis and urinary leakage in a single MUC category. After pooling the data, there was no significant difference between groups in the risk of developing major urological complications (OR 1.01, CI 95%, 0.45 to 2.27, *p* = 0.98) (Figure A3, Appendix A). However, we think that ureteral stenosis and urinary leakage are fundamentally different because these complications have a different pathophysiology, so we should be careful with interpretation of these combined outcome parameters.

### 3.4. Reintervention

Yuksel et al. [23] described the incidence of surgical reintervention because of urological complications after renal transplantation at four different timings of stent removal. There was a clear difference between early (less than three weeks) and late (more than three weeks) stent removal (6.3% versus 1.3%). Patel et al. [21] reported three cases (3.7%) of major urological complications that required surgical revision in the early (five days) versus one case (0.8%) in the late (28 days) stent removal group. Indu et al. [31] reported one case (2.0%) of urinary leakage that required percutaneous nephrostomy in the early stent removal group. Huang et al. [33] reported two cases (1.1%) of urinary leakage that required surgical revision in both the early and late stent removal groups.

Verma et al. [32] reported zero surgical reintervention after major urological complications in both early and late stent removal group (two and four weeks, respectively).

Soldano et al. [35] and Liu et al. [20] investigated surgical reimplantation of the JJ-stent; Soldano et al. [35] reported one case (2.1%) of surgical reimplantation of the stent in the late stent removal group (six weeks), whereas Liu et al. did not report any reimplantation in both the early and late stent removal groups (one and four weeks, respectively). 

## 4. Discussion

There is good evidence that stenting the UNC at the time of KTx is beneficial to reduce major urological complications. Intuitively, transplant professionals feel that ureteric stents should not be in situ for too long to reduce the incidence of infectious complications. However, the optimal timing of stent removal remains unclear. Previous studies show a wide range in the timing of stent removal after KTx (five days until 60 days) [42,43,44,45]. It is already known that the incidence of UTIs is higher when a stent is removed later than five weeks (24.6% to 44%) [13,21,22,34,35]. A UTI after KTx is associated with graft loss, higher morbidity rates, increased risk of rejection and increased hospitalisation rates [6,7,46,47,48]. For this reason, studies have been performed to investigate whether earlier stent removal (e.g., around three weeks) would reduce the incidence of UTIs [8,13,19,20,22,23,31,33,36,37]. We decided to perform a meta-analysis to further investigate if we could define a more optimal timing of stent removal. 

Based on the results, we demonstrated that earlier (<three weeks) stent removal shows a significantly lower incidence of UTIs compared to later removal (>three weeks). Furthermore, earlier stent removal does not appear to lead to a higher incidence of urine leakage. Regarding ureteral stenosis and reintervention, no hard statements can be made, since we were not able to meta-analyse the results. However, overall, incidence or ureteral stenosis and reintervention is clearly very low in kidney transplant recipients (~1 and 3%).

### 4.1. Difficulty in Anastomosis

We realize that characteristics of the study population of both donors and recipients varies and that these are factors that can influence the difficulty of the ureteral anastomosis, and therefore, the outcome after KTx. Unfortunately, only a few studies describe the donor and recipient characteristics in detail; type of donor (living/deceased), type of deceased donor (donor after brainstem/circulatory death), pre-emptive status of the recipients and duration of dialysis are often not given.

Almost every study recorded whether they included a living and/or deceased donor. The majority of the included studies only involved living donor KTx [19,20,22,23,31,32,36,37]; Huang et al. [33] only included deceased donor kidneys. Three studies included both deceased and living donors [8,13,21]. Two studies did not record whether they included living and/or deceased donors [34,35]. 

Furthermore, Patel et al. [21] reported that of the deceased donors, the majority was a DBD—in the early stent group 81.2% and in the late stent group 64.4%. They recorded that in both the early and late stent removal group, 72% of the transplant patients were dialysis dependant before KTx. In the study by Huang et al. [33], all patients were receiving dialysis before KTx. In both groups, around 95% of the transplant patients underwent haemodialysis and 5% peritoneal dialysis. In the early and late stent removal groups, patients were respectively 25.7 and 24.8 months on dialysis prior to transplantation. Coskun et al. [13] only reported that duration of dialysis prior to transplantation varied between 1 and 168 months.

### 4.2. Urinary Tract Infection

The incidence of UTIs varied widely between included studies: 0 to 53% [8,13,19,20,21,22,23,31,32,33,34,35,36,37]. Most transplant centres used a similar triple-regimen immunosuppressive therapy, consisting of calcineurin inhibitors (tacrolimus or cyclosporine), mycophenolate mofetil and corticosteroids. For all details of the immunosuppressive therapy see Table A1 in appendix A. We noticed that some studies did not describe the immunosuppressive drugs nor the prophylactic antibiotics used. We assumed that in these cases, a calcineurin inhibitor-based triple-immunosuppressive regimen was used, and antibiotic prophylaxis was prescribed. Yuksel et al. [23] attribute the low incidence of UTIs to their strict regime of antibiotic prophylaxis. In addition, Sarier et al. [22] and Wilson et al. [6] stress the importance of prophylactic co-trimoxazole to protect against UTIs after transplantation. Furthermore, they state that previous in vivo and in vitro studies have demonstrated that the antibiotic types fluoroquinolones, aminoglycosides and beta-lactam antibiotics may be effective in prevention of the biofilm mechanism—a major problem in bacterial stent colonization [22]. 

Coskun et al. [13] state that UTIs should rather be treated with earlier stent removal opposed to the prescription of antibiotics. We agree with this statement and would advise transplant centres to remove the (potential) source of UTIs earlier rather than later as a best possible way to prevent UTIs. 

Some studies investigated very early stent removal, around one week post-transplantation, and it showed promising results, specifically (and maybe only) regarding the incidence of UTIs [20,21,23,31,34,36].

Dadkhah et al. [37] showed a remarkably high incidence of UTIs in the early stent removal group (ten days), which was two times higher than for late stent removal (30 days). Surprisingly, they concluded that removal of the ureteral stent shortly after KTx has a statistically negligible impact on the rate of UTIs. We decided to include their controversial incidence of UTIs in our meta-analysis. However, one should keep in mind that this study had some paucity of data granularity, as the technique of UNC, length of follow up, immunosuppressive regimen, the use of antibiotic prophylaxis and technique of stent removal were not described. 

### 4.3. Ureteral Leakage and Ureteral Stenosis

Included studies with wide varying timings of stent removal show that ureteral leakage and stenosis are complications with low incidences (0–3%). Three studies described a remarkably high incidence of urinary leakage [32,35,38]. Asgari et al. [38] reported 6.6% urinary leakage in the early stent removal group (ten days) and 13.3% in the late stent removal group (30 days). Soldano et al. [35] reported 6.3% urinary leakage in the late stent removal group (at six weeks). Verma et al. [35] reported 5.8% and 10% urinary leakage in the early (two weeks) and late (four weeks) stent removal groups. Because these remarkably high incidences were derived from (pilot) studies with small patient populations (each group containing respectively around 50 patients), we have to interpret these data very carefully.

Overall, the data implies that both leakage and stenosis can be successfully prevented with insertion of a stent but the duration of the stent being in situ does not have a great influence on the incidence of these urological complications [5,6,10,11,12]. Huang et al. [33] support this by concluding that stent removal at three weeks is as effective in preventing urological complications as removal at six weeks (with similar prophylactic antibiotics and immunosuppressive therapies). Furthermore, Patel et al. [21] demonstrate that very early stent removal (less than five days) results in a lower incidence of leakage and stenosis than in the un-stented population. Therefore, even when the stent is inserted for a brief period, it already shows benefit in preventing leakage and stenosis. The first two weeks after KTx are believed to have the highest incidence of urinary leakage and ureteral stenosis [8,13,23,32]. Yuksel et al. [23] conclude that stent removal earlier than fourteen days shows a significant increase of recurrent surgical UNC intervention. In addition, Coskun et al. [13] conclude that stent removal at two weeks results in acceptable mucosal healing of the anastomosis to prevent urological complications. In order to keep the incidence of urinary leakage and ureteral stenosis as low as possible, we recommend that stents should not be removed earlier than two weeks.

### 4.4. Additional Advantages

In addition to a lower incidence of UTIs, early stent removal has other advantages. Instead of a cystoscopy, stents can be removed less invasively together with the removal of the urinary catheter if tied to it at the time of transplantation. This procedure is considered far more comfortable for the patient [8,20]. Additionally, early stent removal provides the opportunity to remove the stent during the same admission, which leads to a reduction in costs and fewer forgotten stents [5,21,32,33].

### 4.5. Limitations

A meta-analysis can only be as good as the quality of the included studies. Unfortunately, most of the included studies are retrospective cohort studies. Only three RCTs were included [20,21,31]. We mention this limitation to alert the reader to carefully interpret the data. In the forest plot analysing urinary tract infections for early (<3 weeks) versus late (>3 weeks) stent removal (Figure 3), one can appreciate the relatively high heterogeneity (*I*^2^ = 61%). The cause of this heterogeneity lies in the differences in study design; for example, not defining which type of stent has been used, technique of stent removal, technique of ureteroneocystostomy, use of prophylactic antibiotics or choice of immunosuppressive therapy. The latter two aspects are particularly important factors influencing UTI rates. Furthermore, urinary leakage and ureteral stenosis were not always defined, or no uniform definition was adhered to. This leads to assumptions, which may cause inadequate comparison. Studies often do not report at what time during follow-up, especially before or after stent removal, specific complications occurred. 

Recently, Thompson et al. [24] also investigated the benefits of early stent removal. They concluded that early stent removal reduces the incidence of UTIs, while it was uncertain if there is a higher risk of MUC. Furthermore, the authors used incidence of MUC as a primary outcome and UTIs as a secondary outcome. In our study, UTIs and MUC were chosen as, respectively, primary and secondary outcome, because our main focus was to investigate whether early stent removal reduced the disadvantages of stenting (incidence of UTIs), without compromising the beneficial effects (preventing MUC).

Thompson et al. [24] classified early and late stent removal in a different manner; although they mention that early stent removal was defined as stent removal below fifteen days, they did not particularly use this definition in their analyses. The authors copied the “early” and “late” groups from included studies. As a consequence, there is no common cut-off value analysed in their study and meta-analysis. 

Furthermore, the focus of their analysis was more on the specific type of stents used (bladder indwelling stent and per-urethral stents). In our opinion, we should first focus on the relation between duration of stenting and the incidence of UTIs rather than the influence of the type of stent.

## 5. Conclusions 

The results of this systematic review clearly point favourably towards an earlier stent removal around three weeks opposed to the six weeks that is currently used in most transplant centres. Earlier stent removal (at or below three weeks) results in fewer UTIs without negatively affecting the anastomosis between ureter and bladder. We recommend at this stage that ureteric stents should not be removed earlier than two weeks. We would recommend initiating an RCT, randomising between very early stent removal at one week and stent removal at three weeks. Another option would be a three-armed RCT, adding an additional group of stent removal at six weeks. 

## Figures and Tables

**Figure 1 jcm-08-00689-f001:**
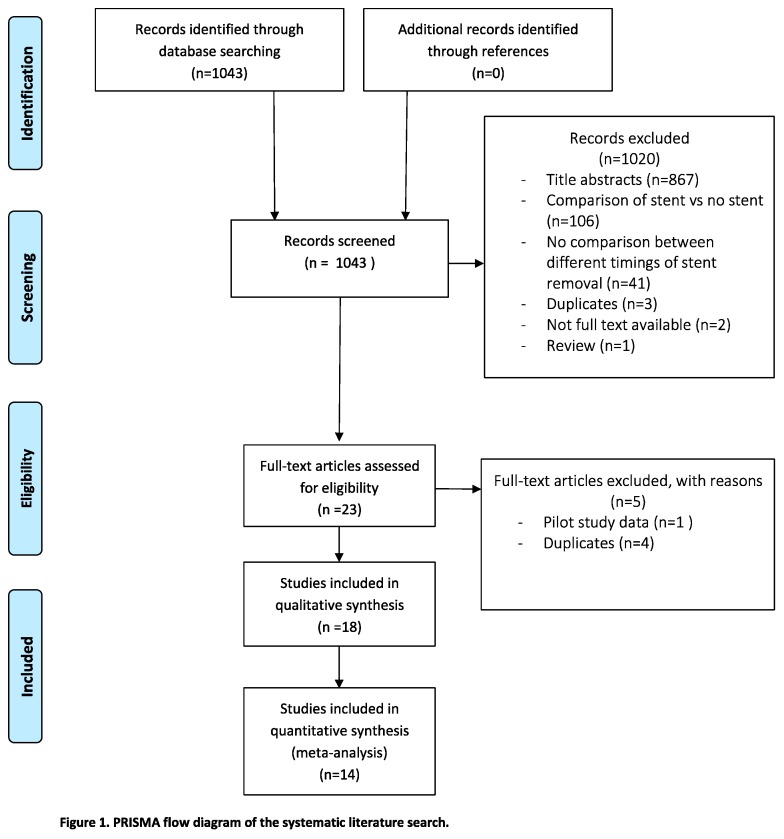
Preferred Reporting Items for Systematic Reviews and Meta-Analyses (PRISMA) flowchart.

**Figure 2 jcm-08-00689-f002:**
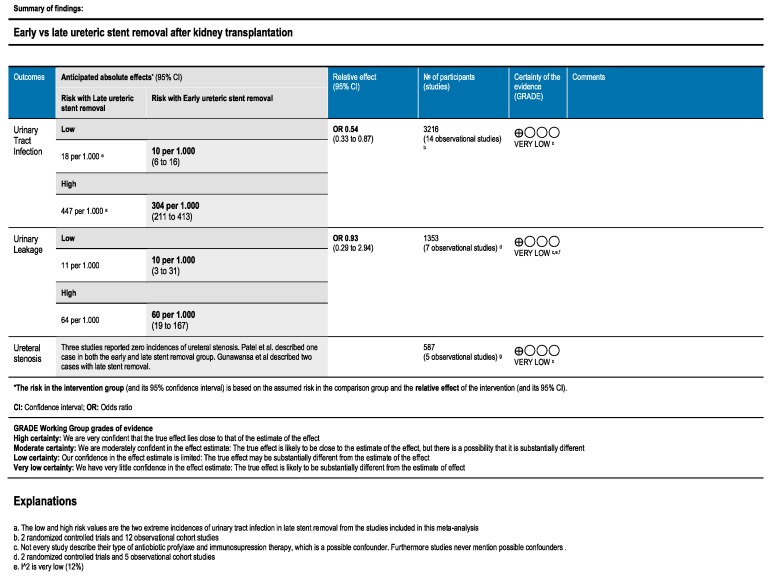
Quality assessment of the included studies.

**Figure 3 jcm-08-00689-f003:**
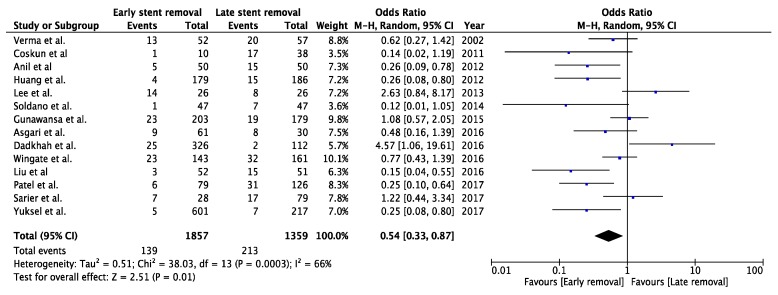
Forest plot of urinary tract infection for early (<3 weeks) versus late (>3 weeks) stent removal.

**Figure 4 jcm-08-00689-f004:**
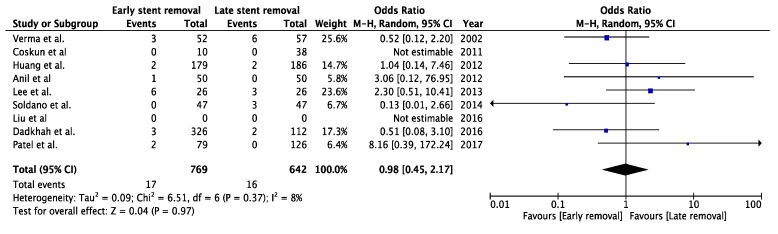
Forest plot urinary leakage for early (<3 weeks) versus late (>3 weeks) stent removal.

**Table 1 jcm-08-00689-t001:** Overview of the included studies.

Studies	Year	Sort Study	Timing of Stent Removal	Number of Patients	Urinary Tract Infection	Major Urological Complication	Urinary Leakage	Ureteral Stenosis
Yuksel et al. [23]	2017	Retrospective	5–7 days	153	x	x		
			8–14 days	165				
			15–21 days	283				
			>22 days	217				
Patel et al. [21]	2017	RCT	5 days	79	x	x	x	x
			6 weeks	126				
Sarier et al. [22]	2017	Retrospective	15–21 days	28	x			
			21–28 days	54				
			28–35 days	25				
Wingate et al. [8]	2017	Retrospective	<3 weeks	143	x			
			>3 weeks	161				
Liu et al. [20]	2016	RCT	7 days	52	x	x	x	x
			28 days	51				
Dadkhah et al. [37]	2016	Prospective	10 days	164	x	x	x	
			20 days	162				
			30 days	112				
Asgari et al. [38]	2016	Retrospective	10 days	30	x	x	x	
			20 days	31				
			30 days	30				
Gunawansa et al. [36]	2014(1)	Prospective	6 days	203	x	x		x
		randomised	28 days	179				
Soldano et al. [35]	2014(2)	Retrospective	5 days	47	x	x	x	
			6 weeks	47				
Lee et al. [34]	2013(3)	Retrospective	5 days	26	x	x		
			6 weeks	26				
Huang et al. [33]	2012	Retrospective	3 weeks	179	x	x	x	x
			6 weeks	186				
Indu et al. [31]	2012	Prospective RCT	7 days	50	x	x	x	
			28 days	50				
Coskun et al. [13]	2011	Retrospective	13–14 days	10	x	x		
			>20 days	38				
Verma et al. [32]	2002	Retrospective	2 weeks	52	x	x	x	x
			4 weeks	57				

RCT: Randomised controlled trial. (1) Abstract of the 17th Congress of the European Society for Organ Transplantation; (2) Abstract of the the World Transplant Congress 2014; and (3) Abstract of the 16th Congress of the European Society for Organ Transplantation.

**Table 2 jcm-08-00689-t002:** Overview of the measured outcome parameters.

Studies	Stent Removal	Number of Patients	Urinary Tract Infection (%)	Major Urological Complication	Ureteral Stenosis (%)	Urinary Leakage (%)	Surgical Reintervention (%)
Yuksel et al. [23]	5–7 days	153	0% *	11.0%			11.0%
	8–14 days	165	1.2% *	9.6%			9.6%
	15–21 days	283	1.1% *	1.7%			1.7%
	>22 days	217	3.2% *	1.3%			1.3%
Patel et al. [21]	5 days	79	7.6% *	3.7%	1.2%	2.5%	3.7%
	6 weeks	126	24.6% *	0.8%	0.8%	0%	0.8%
Sarier et al. [22]	15–21 days	28	7.1% *	x			
	21–28 days	54	5.6% *	x			
	28–35 days	25	12.0% *	x			
Wingate et al. [8]	<3 weeks	143	31.7% *	x			
	>3 weeks	161	51.6% *	x			
Liu et al. [20]	7 days	52	5.8% *	0%	0%	0%	0%
	28 days	51	29.4% *	0%	0%	0%	0%
Dadkhah et al. [37]	10 days	164	18.1%	1.0%		1.0%	
	20 days	162	5.7%	1.0%		1.0%	
	30 days	112	9.1%	2.8%		2.8%	
Asgari et al. [38]	10 days	30	20.0%	6.6%		6.6%	
	20 days	31	9.7%	6.4%		6.4%	
	30 days	30	26.7%	13.3%		13.3%	
Gunawansa et al. [36]	6 days	203	11.3%	0%	0%		
	28 days	179	10.6%	1.1%	1.1%		
Soldano et al. [35]	5 days	47	10.6% *	0%		0%	0%
	6 weeks	47	25.5% *	6.3%		6.3%	2.1%
Lee et al. [34]	5 days	26	53.0%	23.0%			
	6 weeks	26	30.0%	12.0%			
Huang et al. [33]	3 weeks	179	2.2% *	1.1%	0%	1.1%	1.1%
	6 weeks	186	8.1% *	1.1%	0%	1.1%	1.1%
Indu et al. [31]	7 days	50	14.0% *	2.0%		2.0%	0%
	28 days	50	38.0% *	0%		0%	0%
Coskun et al. [13]	13–14 days	10	10.0% *	0%			
	>20 days	38	45.0% *	0%			
Verma et al. [32]	2 weeks	52	25.0% *	5.8%	0%	5.8%	0%
	4 weeks	57	35.1% *	10.0%	0%	10.0%	0%

* Defined as urinary tract infection (UTI).

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
