# Peer review of "Timing of Ureteric Stent Removal and Occurrence of Urological Complications after Kidney Transplantation: A Systematic Review and Meta-Analysis"

_jcm, 2019, doi:10.3390/jcm8050689_

Reviewer 1 Report

These authors perform a systematic review and meta-analysis to compare the efficacy and safety of early vs late ureteric stent withdrawal after kidney transplantation. The concept is not novel, and we already know that the earlier the stent withdrawal, the lower the urinary tract infection rates, without increasing urinary leaks or stenoses. 

The 14 studies selected for the analyses are very different in quality, with very few randomized trials and many retrospective series. I am not sure that this analysis adds any relevant knowledge to the previous Cohrane metaanalysis (Ref 48) or, for instance, the RCT from Patel et al AJT 2017. 

Author Response

Our response to the reviewers's comments can be found in the rebuttal letter in the word file below.

Author Respons:

Point 1: These authors perform a systematic review and meta-analysis to compare the efficacy and safety of early vs late ureteric stent withdrawal after kidney transplantation. The concept is not novel, and we already know that the earlier the stent withdrawal, the lower the urinary tract infection rates, without increasing urinary leaks or stenoses. 

The reviewer is absolutely right that the concept is not novel, and we do not wish to suggest that either. The aim of this systematic review and meta-analysis is to generate as robust evidence as possible to verify whether this approach is justified; many centres still remove ureteric stents at much later timepoints (even >6 weeks), and certainly guidelines have not all adopted early removal of ureteric stents after KTx in their recommendations.

Point 2: The 14 studies selected for the analyses are very different in quality, with very few randomized trials and many retrospective series. I am not sure that this analysis adds any relevant knowledge to the previous Cohrane meta-analysis (Ref 48) or, for instance, the RCT from Patel et al AJT 2017. 

We fully agree with the reviewer that it is disappointing that the quality of the included studies is fairly low, and that the studies largely consisted of retrospective studies, and only three RCTs. We fully acknowledge this in the Limitations section. Unfortunately, this is the nature of systematic reviews and meta-analyses; we have to work with whatever is published. We did not want to exclude studies based on their quality, as this would lead to significant bias.

When we started our study, the Cochrane review and meta-analysis had been registered in the Prospero database for two years and did not show any progression in those years, even now it is still registered in the stage “piloting of the study process”. We therefore decided to go ahead with our study and we were somewhat surprised when their paper got published.

Furthermore, compared to the Cochrane meta-analysis, we have included more studies (n= 14 versus n=5), more patients (n= 3216 versus n=1127) and have come to more robust conclusions with regards to timing of stent removal, as they did not have a common cut-off value for early versus late stent removal. In addition, they were more interested in the difference between per-urethral and bladder indwelling stents, while we focused on all types of stents. As we addressed in our discussion, we think that we should first focus on the relation between duration of stenting and the incidence of urological complications rather than the influence of type of stent.       

Patel’s RCT published in AJT 2017 included a relatively small number of patients compared to the number of patients included in our study (n=205 vs n=3216), with very low events (especially for urinary leakage and ureteral stenosis). In our opinion, conclusions based on the Patel RCT are less strong than those of a well-executed systematic review and meta-analysis (despite the low-quality studies included).         

We think it is actually great news that the two meta-analyses are in concordance in terms of conclusions, which should stimulate new/revised guidelines to include these findings without hesitation as Level 1A evidence. Currently, to our knowledge, only one guideline (EAU) includes an advice on timing of ureteric stent removal.

Reviewer 2 Report

Overall, this is a well written paper. Since this is a meta-analysis, originality is not the issue. The issue of stent vs. no stent and if a stent the timing of removal have been the subject on an ongoing dialogue for a number of years.

The quality of studies that address the timing of stent removal are modest at best. That is the principal problem with the meta-analyses of such studies.

However, given the limitations, the conclusion and the general direction seems reasonable and pragmatic. Stent removal around a 3 week time after kidney transplantation seems very balanced from the perspective of UTI and leaks.

The authors do acknowledge that immunosuppression data were not readily available as that is one other variable for early infections. Another variable was early rejections and using anti lymphocyte therapy could increase the risk of infections.

Other complications such as ureteric necrosis/rejection could lead to urinary leak, and may not necessarily be related to the stent removal. However, those things are beyond the scope of this paper.

Kidney transplant might be the best option/choice for ESRD management. Perhaps, "golden standard" could be rephrased that way!

Author Response

Point 1: Overall, this is a well written paper. Since this is a meta-analysis, originality is not the issue. The issue of stent vs. no stent and if a stent the timing of removal has been the subject on an ongoing dialogue for a number of years.

We thank the reviewer for the positive feedback.

Point 2: The quality of studies that address the timing of stent removal are modest at best. That is the principal problem with the meta-analyses of such studies.

We agree with the reviewer about the above-mentioned point, and we have elaborated this is our answer to reviewer 1. 

Point 3: However, given the limitations, the conclusion and the general direction seems reasonable and pragmatic. Stent removal around a 3-week time after kidney transplantation seems very balanced from the perspective of UTI and leaks.

We thank the reviewer for the positive feedback and agree on this.

Point 4: The authors do acknowledge that immunosuppression data were not readily available as that is one other variable for early infections. Another variable was early rejections and using anti lymphocyte therapy could increase the risk of infections.

We think indeed that this highlights that we as a community should do better in terms of reporting relevant information like this in our clinical studies. One of the advantages of a systematic review and meta-analysis is, that we actually can identify these issues, highlight them to the readership to increase awareness and enable colleagues to carefully interpret data coming from these studies.

Point 5: Other complications such as ureteric necrosis/rejection could lead to urinary leak, and may not necessarily be related to the stent removal. However, those things are beyond the scope of this paper.

Agreed.

Point 6: Kidney transplant might be the best option/choice for ESRD management. Perhaps, "golden standard" could be rephrased that way!

Many thanks for this suggestion; we have changed this accordingly in the MS.

We feel that your feedback helped improve our manuscript and we hope that our manuscript is now acceptable for publication in the Journal of Clinical Medicine. We look forward to hearing from you. If you have any further comments or questions, please do not hesitate to contact us.

Kind regards,

Round  2

Reviewer 1 Report

The systematic review is correct, the methodology is correct, I simply feel that this meta-analysis does not add any relevant information on top of the existing knowledge. The conclusions are confirmatory in nature and we already know that ureteral stent should be removed as soon as possible after kidney transplantation. As may be read in the authors' responses to these comments, they agree, but justify it: "not new knowledge, but useful".